# FEDOPENMATCH: TOWARDS SEMI-SUPERVISED FEDERATED LEARNING IN OPEN-SET ENVIRONMENTS

**Hongquan Liu**[1]**, Chenyu Guo**[1]**, Yixin Ren**[1]**, Jihong Guan**[2*]**, Shuigeng Zhou**[1*]

[1]Shanghai Key Lab of Intelligent Information Processing, and
 College of Computer Science and Artificial Intelligence, Fudan University, Shanghai, China
[2]School of Computer Science and Technology, Tongji University, Shanghai, China
`{hqliu21,guocy24,yxren21}@m.fudan.edu.cn`
`jhguan@tongji.edu.cn, sgzhou@fudan.edu.cn`

## ABSTRACT

Semi-supervised federated learning (SSFL) has emerged as an effective approach to leverage unlabeled data distributed across multiple data owners for improving model generalization. Existing SSFL methods typically assume that labeled and unlabeled data share the same label space. However, in realistic federated scenarios, unlabeled data often contain categories absent from the labeled set, i.e., outliers, which can severely degrade the performance of SSFL algorithms. In this paper, we address this under-explored issue, formally propose the open-set semi-supervised federated learning (OSSFL) problem, and develop the first OSSFL framework, FedOpenMatch. Our method adopts a one-vs-all (OVA) classifier as the outlier detector, equipped with logit adjustment to mitigate inlier-outlier imbalance and a gradient stop mechanism to reduce feature interference between the OVA and inlier classifiers. In addition, we introduce the logit consistency regularization loss, yielding more robust performance. Extensive experiments on standard benchmarks across diverse data settings demonstrate the effectiveness of FedOpenMatch, which significantly outperforms the baselines.

## 1 INTRODUCTION

Federated learning (FL) allows multiple clients to collaboratively train models without sharing private data (McMahan et al., 2017; Kairouz et al., 2021). The privacy preserving nature has driven its wide applications in various fields, such as medicine (Sheller et al., 2020) and autonomous driving (Eid Kishawy et al., 2024). However, typical FL assumes that all distributed training samples are fully labeled, which is impractical in most cases as it requires that all clients have enough expertise knowledge, time and willingness to label their data.

Recently, *semi-supervised federated learning* (SSFL) has emerged as an effective solution to exploit the distributed unlabeled data with accessing to a limited number of labeled samples (Jeong et al., 2021). SSFL generates pseudo-labels for unlabeled data and then involves them into training. SSFL works can be approximately categorized into two types, *label-at-client* (Liu et al., 2024; Bai et al., 2024) and *label-at-server* (Diao et al., 2022; Lee et al., 2024), with respect to the location of labeled data. This work focuses on the latter setting, which is considered more practical as it does not assume clients have labeling capabilities. Instead, the server (e.g., autonomous driving company) often has plenty resources, and can afford to acquire a small number of labeled samples at a reasonable cost.

While previous works have made great progress in utilizing the unlabeled data, they all assume that unlabeled data share the same category set with labeled data. However, as the clients collect training samples independently and privately, it is inevitable for them to contain samples from unseen classes, i.e., outliers, so as the testing data. Standard SSFL algorithms do not have the ability to detect outliers and they will assign wrong pseudo-labels for outliers, which has been proven to hurt the model performance (Oliver et al., 2018). Moreover, outliers will be wrongly predicted as seen classes at inference stage, causing inevitable failures in critical application scenarios. Thus, it is critical to take the existence of outliers into consideration when designing SSFL algorithms.

---

*Corresponding author

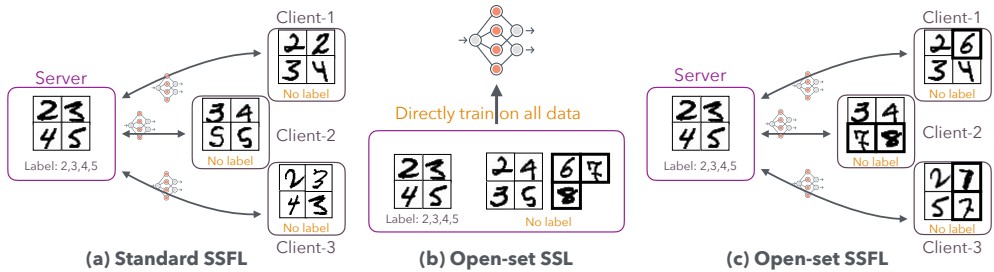

Figure 1: Illustration of the differences among (a) standard SSFL where all data are in the same category space ('2', '3', '4', '5'), (b) open-set SSL (OSSL) where the unlabeled data contain samples of unseen classes ('6', '7', '8'), and (c) our open-set SSFL (OSSFL) where unlabeled data in the clients have samples of classes ('6', '7', '8') different from that of the labeled data in the server.

On the other hand, there is a related research field called *open-set semi-supervised learning (OSSL)* (Saito et al., 2021), which is for centralized scenarios, where the model can access both labeled and unlabeled data simultaneously, and the unlabeled data contain samples of unseen classes. While federated settings pose additional challenges: the strict separation of labeled and unlabeled data makes client difficult to obtain correct supervision, and client training can be easily misled by noisy pseudo-labels. Moreover, data heterogeneity across clients further exacerbates this issue. Therefore, naively adapting those OSSL algorithms into the federated learning setting will suffer severe performance degradation, as detailed in Sec. 4.2.

To address the problem caused by outliers in SSFL, in this paper we first formulate a new problem called Open-set Semi-supervised Federated Learning (OSSFL in short), in other words, OSSL in FL. Fig. 1 illustrates the differences among SSFL, OSSL and our OSSFL. Concretely, SSFL is SSL in FL setting; OSSL is SSL in centralized open-set scenarios; and our OSSFL is SSL in federated open-set scenarios. To the best of our knowledge, up to now there has been no OSSFL work in the literature. Then, we design the first OSSFL framework, **FedOpenMatch**. Specifically, we employ an *one-vs-all (OVA)* classifier serving as the outlier detector to help generate high-quality inlier pseudo-labels during training. We also introduce gradient stop to address the interference issue between inlier and OVA classifier, and employ logit adjustment for combating the imbalanced OVA prediction, avoiding the inliers being recognized as outliers and discarded during training. To further exploit the unlabeled samples including both unconfident inliers and outliers, we devise a logit consistency regularization loss by modifying the early proposed probability consistency regularization loss (Saito et al., 2021), which significantly improves the performance. Comprehensive experiments on three commonly used benchmarks under various data settings show that FedOpenMatch can significantly boost the open-set accuracy up to 14.33% on the CIFAR-100 dataset.

In summary, the contributions of this paper are as follows: 1) We formally propose the new problem: open-set semi-supervised federated learning (OSSFL), to tackle the outlier issue in SSFL, i.e., to extend OSSL to FL scenarios. 2) We develop a simple but powerful OSSFL framework, FedOpenMatch, which consists of three components: gradient stop, logit adjustment, and logit consistency regularization. 3) For comprehensive performance evaluation and comparison, we implement several OSSFL baselines by adapting the latest OSSL algorithms to the OSSFL setting. 4) We conduct extensive experiments to evaluate FedOpenMatch, which show that FedOpenMatch significantly outperforms the baselines on commonly used benchmarks across diverse data settings.

## 2 RELATED WORK

**Semi-supervised Learning (SSL).** SSL is a learning paradigm that leverages both labeled and unlabeled data. Early advances in SSL were largely driven by two techniques: consistency regularization (Xie et al., 2020; Miyato et al., 2018) and pseudo-labeling (Lee et al., 2013). More recently, FixMatch (Sohn et al., 2020) integrates both strategies into a simple yet highly effective framework, achieving high performance across a variety of benchmarks. Building on this foundation, subsequent works have introduced further refinements. For example, FlexMatch (Zhang et al., 2021), FreeMatch (Wang et al., 2022), InstanT (Li et al., 2023a), and Park et al. (2025) adopt fine-grained

Table 1: A qualitative comparison among SSFL, OSSL, FOSR, and OSSFL from four perspectives.

| | Distributed | Scarce labels | Open-set unlabeled training | Open-set testing |
|---|---|---|---|---|
| **FOSR** | ✓ | ✗ | ✗ | ✓ |
| **SSFL** | ✓ | ✓ | ✗ | ✗ |
| **OSSL** | ✗ | ✓ | ✓ | ✓ |
| **OSSFL** | ✓ | ✓ | ✓ | ✓ |

thresholding mechanisms to generate high-quality pseudo-labels. SoftMatch (Chen et al., 2023) proposes a sample re-weighting strategy, while DST (Chen et al., 2022) and FlatMatch (Huang et al., 2023) enhance robustness against noisy pseudo-labels. Despite their success, all of these methods share a fundamental limitation: they assume that labeled and unlabeled data are drawn from the same category set, which rarely holds in real-world scenarios.

**Open-set SSL.** This line of research extends conventional SSL to scenarios where unlabeled data may contain samples from unseen classes. Early OSSL methods typically follow a *detect-and-filter* paradigm, aiming to reduce the negative impact of outliers by detecting and discarding them. Different approaches have been explored for outlier detection, including ensemble confidence (Chen et al., 2020), energy-based scoring function (He et al., 2022), binary classification (Yu et al., 2020), weighting function (Guo et al., 2020), and cross-modal matching (Huang et al., 2021). More recent methods achieve higher performance by employing *one-vs-all (OVA)* classifiers as outlier detectors (Saito et al., 2021; Li et al., 2023b; Fan et al., 2023; Hang & Zhang, 2024), along with auxiliary strategies such as probability consistency regularization (Saito et al., 2021) and negative pseudo-label mining (Fan et al., 2023; Hang & Zhang, 2024). To reduce the dependence on a reliable OVA classifier, IOMatch (Li et al., 2023b) reformulates OSSL into a $(K + 1)$-way classification problem by grouping all unseen categories into a single "unknown" class. In contrast, BDMatch (Hang & Zhang, 2024) redefines OSSL as multiple binary SSL tasks, thereby unifying the pseudo-labeling space for both seen and unseen samples.

**Semi-supervised Federated Learning.** Existing SSFL works can be broadly categorized into two paradigms: *label-at-client*, where each client has access to both labeled and unlabeled samples (Liang et al., 2022; Zhang et al., 2023a;b; Bai et al., 2024; Liu et al., 2024; Zhang et al., 2024; Liao et al., 2025); and *label-at-server*, where clients hold only unlabeled data while the server maintains a small labeled dataset (Jeong et al., 2021; Diao et al., 2022; Kim et al., 2023; Yang et al., 2024; Lee et al., 2024). This work focuses on the *label-at-server* setting, which is considered more realistic in practice. A key challenge in SSFL is the generation of reliable pseudo-labels to maximize the utility of distributed unlabeled data. Several strategies have been proposed. FedMatch (Jeong et al., 2021) adopts disjoint learning to reduce interference between labeled and unlabeled knowledge. SemiFL (Diao et al., 2022) mitigates pseudo-label degradation by generating pseudo-labels with global model instead of local ones. $FL^2$ (Lee et al., 2024) introduces sharpness-aware consistency regularization and adaptive thresholding, enabling more effective exploitation of unlabeled data. Kim et al. (2023) and Yang et al. (2024) focus on different tasks.

**Federated Open-set Recognition (FOSR).** FOSR (Yang et al., 2023; Zhu et al., 2023) has the same goal to accurately classify samples from seen classes while detecting outliers from unseen classes, but differs in the training process. FOSR trains models with completely labeled samples from seen classes while OSSFL trains with unlabeled open-set dataset. This distinction makes OSSFL more challenging, as it must tackle the difficulties of semi-supervised learning and open-set recognition.

**Summary.** To discriminate this paper with existing works, Tab. 1 presents a qualitative comparison between OSSFL and three related research areas SSFL, OSSL and FOSR from four perspectives. Existing SSFL methods are vulnerable to outliers during training and fail to identify outliers during testing. Although Zhang et al. (2023b) introduces an open-world SSFL framework, it is limited to the label-at-client setting and pursues a different objective than open-set SSFL. OSSL methods share the same goal as OSSFL, but they cannot be directly adopted in the federated setting due to the strict separation of labeled and unlabeled data and the data heterogeneity across clients. These limitations motivate us to propose the OSSFL problem and develop the FedOpenMatch framework.

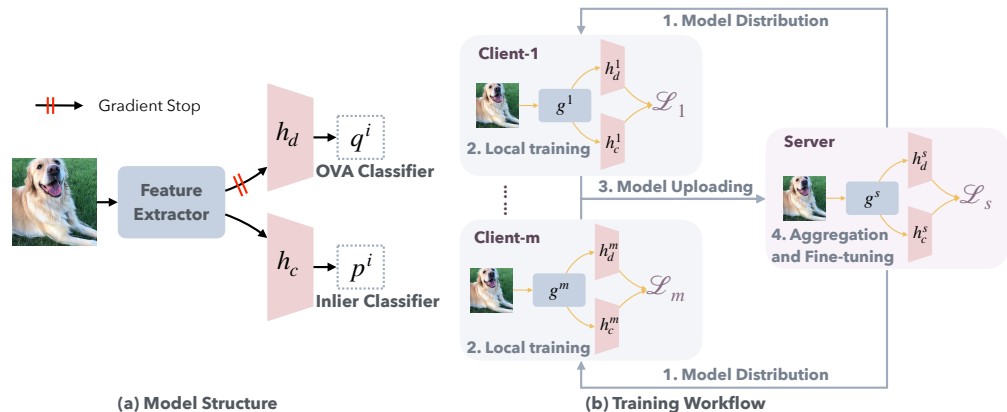

Figure 2: (**a**) The model architecture, consisting of a shared feature extractor, an inlier classifier, and an OVA classifier. (**b**) The training workflow. In each communication round, the server and clients alternatively update their models, which is abstracted into four stages: (1) model distribution, (2) client training, (3) model uploading, and (4) server aggregation and fine-tuning.

## 3 THE FEDOPENMATCH METHOD

### 3.1 PROBLEM DEFINITION AND FRAMEWORK

**Problem definition.** We assume there are a central *server* holds a small labeled dataset $D_s = \{x_{s;i}, y_{s;i}\}_{i=1}^{N_s}$ where $y_{s;i} \in Y = [1, \ldots, K]$ is the label of sample $x_{s;i}$ and $Y$ is the set of indexes of $K$ seen classes, and $M$ *clients*, each of which has only an unlabeled dataset $D_u^m = \{x_{m;i}\}_{i=1}^{N_m}$. Typically, $\sum_{m=1}^{M} N_m \gg N_s$. In the open-set setting, the category of unlabeled sample $x_{m;i}$ is not guaranteed to be one of the $K$ seen classes. The goal of OSSFL is to learn a model that can classify the unlabeled samples of seen classes (i.e., inliers) into correct seen classes while identifying the unlabeled samples of unseen classes (i.e., outliers).

**Framework.** FedOpenMatch is a multi-task learning framework that jointly trains an inlier classifier and an outlier detector. The whole model $f$ consists of a shared feature extractor $g$, an inlier classifier $h_c$ and an OVA classifier $h_d$. For the $K$-classification task, the inlier classifier outputs a $K$-dimensional logit $p : \{p^k\}_{k=1}^K = h_c \circ g(x)$. The OVA classifier is composed of $K$ binary classifiers, each distinguishing the $k$-th target class from the remaining seen and unseen classes. For a sample $x$, it outputs $2K$-dimensional logit $q : \{q^k : \{q_0^k, q_1^k\}\}_{k=1}^K = h_d \circ g(x)$, where $q_0^k$ and $q_1^k$ represent the score of $x$ being an outlier or an inlier with respect to class $k$. For simplicity, we denote by $\hat{p}_i$ and $p_i$ the inlier predictions of $\alpha(x_i)$ and $\mathcal{A}(x_i)$ respectively. Here, $\alpha(\cdot)$ and $\mathcal{A}(\cdot)$ denote weak data augmentation (e.g., random cropping and flipping) and strong data augmentation (RandAugment (Cubuk et al., 2020)). Similarly, $\hat{q}_i$ and $q_i$ denote the OVA predictions.

In each communication round $t$, the server first trains the model by minimizing

$$\mathcal{L}_s = \frac{1}{N_s} \sum_{i=1}^{N_s} \ell_{ce}(p_i, y_i) + \ell_{ova}(q_i, y_i), \tag{1}$$

where $\ell_{ce}$ denotes the cross-entropy loss, and $\ell_{ova}$ is the loss of the OVA classifier where we adopt the hard-negative sub-classifier sampling technique (Saito et al., 2021), which is formulated as

$$\ell_{ova}(q_i, y_i) = -\log(q_{i;1}^y) - \min_{k \neq y} \log(q_{i;0}^k). \tag{2}$$

Once the server training is completed, the updated model is distributed to a subset of randomly-selected clients. Then, each selected client $m$ updates the model by optimizing

$$\mathcal{L}_m = \mathcal{L}_m^{in} + \mathcal{L}_m^{ova} + \mathcal{L}_m^{lcr}, \tag{3}$$

where $\mathcal{L}_m^{lcr}$ is the weak-strong logit consistency regularization defined in Eq. (8), $\mathcal{L}_m^{in}$ is the loss of the inlier classifier with confident inlier pseudo-labels, and $L_m^{ova}$ is the loss of the OVA classifier.

Specifically, $\mathcal{L}_m^{in}$ and $L_m^{ova}$ are formulated as follows:

$$\mathcal{L}_m^{in} = \frac{1}{N_m} \sum_{i=1}^{N_m} (\mathbb{I}(\Gamma(\hat{p}_i) \geq \tau_{in}) \wedge \mathbb{I}(\hat{q}_{i;1}^{\hat{y}_i} \geq 0.5)) \ell_{ce}(p_i, \hat{y}_i), \tag{4}$$

$$\mathcal{L}_m^{ova} = \frac{1}{N_m} \sum_{i=1}^{N_m} \sum_{k=1}^{K} (\mathbb{I}(\hat{q}_{i;1}^k \geq \tau_{pos}) \vee \mathbb{I}(\hat{q}_{i;1}^k \leq \tau_{neg})) \ell_{bce}(q_i^k, \mathbb{I}(\hat{q}_{i;1}^k > \hat{q}_{i;0}^k)), \tag{5}$$

where $\hat{y}^i = \arg\max(\hat{p}^i)$ is the inlier pseudo-label, $\Gamma(\hat{p}_i) = \max \operatorname{softmax}(\hat{p}_i)$ computes the confidence of inlier prediction, $\tau_{in}$, $\tau_{pos}$ and $\tau_{neg}$ are predefined thresholds for inlier pseudo-labels, positive and negative OVA pseudo-labels respectively, and $\ell_{bce}$ is the binary cross-entropy loss. After local updates, the client models are aggregated to obtain the new global model $f = \sum_{m=1}^{\mu M} w_m \cdot f_m$, where $\{w_1, \ldots, w_{\mu M}\}$ denotes the aggregation weights, and $\mu$ presents client join ratio.

To ensure pseudo-label stability during local training, we generate pseudo-labels using the global model at the beginning of each round and keep them fixed throughout the client updates. This prevents local training from being biased by noisy labels arising from the scarcity of supervision. Furthermore, we introduce a gradient stop strategy (Sec. 3.2) and a logit adjustment mechanism (Sec. 3.3) to further improve the performance.

## 3.2 STOP GRADIENT FOR DECOUPLED FEATURE SPACE

In our framework, the OVA classifier and the inlier classifier share the same feature extractor, but differ in optimization objectives. The inlier classifier pulls features toward their corresponding class centers while pushing them away from that of the other classes. In contrast, the OVA classifier seeks to separate each target class from the remaining classes. Consequently, the two branches inevitably interfere with each other in the shared feature space. Prior OSSL methods (Fan et al., 2023; Li et al., 2023b) have attempted to mitigate this issue by introducing projection layers to map features into task-specific spaces. However, the divergence in update directions persists, as evidenced by the low similarity between feature gradients of the two branches (see the results in Fig. 5 of Appendix A.2).

To address this challenge, we propose gradient stop (GS), blocking the gradient flow from the OVA branch to the shared feature extractor. The intuition is that the OVA classifier can effectively distinguish inliers from outliers using a feature space where intra-class compactness and inter-class separability are already enforced by the inlier classifier. Our ablation studies (Sec. 4.3) further validate the effectiveness of this gradient stop strategy.

## 3.3 LOGIT ADJUSTMENT FOR BALANCED OVA PREDICTION

During training, we observe that most unlabeled samples (including true inliers) are classified as outliers and thus discarded, leading to a low data utilization rate (see Fig. 10 in Appendix A.2). This phenomenon arises because the OVA classifier is inherently biased toward the majority class—outliers. Specifically, for each class $k$, all samples of the remaining $K$-1 classes serve as outliers during training. As $K$ grows, the imbalance between inliers and outliers becomes more severe, which biases the binary classifiers toward predicting outliers, causing many true inliers to be mistakenly rejected.

To address this, we adopt a logit adjustment strategy inspired by Menon et al. (2021). Specifically, the logits for the OVA loss $\ell_{ova}$ defined in Eq. (2) are corrected as follows:

$$q^k = q^k + \omega \log \pi, \tag{6}$$

where $\pi = \{\frac{K-1}{K}, \frac{1}{K}\}$ represents the class prior probabilities of outliers and inliers, under the assumption of a balanced labeled dataset, and $\omega$ is a tunable scaling factor. This adjustment counteracts the imbalance by amplifying the contribution of inlier predictions, preventing them from being overwhelmed by the dominant outlier updates. We further validate the effectiveness of this logit adjustment in Sec. 4.3.

## 3.4 WEAK-STRONG LOGIT CONSISTENCY REGULARIZATION

In semi-supervised learning, effectively exploiting unlabeled samples is crucial for improving model performance. In Sec. 3.3, we introduce logit adjustment to ensure that as many inliers as possible

are utilized. Nevertheless, a large portion of unlabeled data—including low-confidence inliers and outliers—still remain underexploited. Prior OSSL work (Saito et al., 2021) proposed the *soft open-set consistency regularization loss* (SOCR), defined as

$$\mathcal{L}^{oc} = \lambda \frac{1}{N_m} \sum_{i=1}^{N_m} mse(\text{softmax}(q_i), \text{softmax}(\hat{q}_i)) + mse(\text{softmax}(p_i), \text{softmax}(\hat{p}_i)), \quad (7)$$

where $\lambda$ is a weighting factor. SOCR encourages the model to focus on label-related features by minimizing the divergence between predictions from two augmented views of the same sample. Interestingly, we find that removing the softmax operation and directly enforcing consistency at the *logit* level yields better performance. We denote this as the *logit consistency regularization* (LCR):

$$\mathcal{L}_m^{lcr} = \lambda \frac{1}{N_m} \sum_{i=1}^{N_m} mse(q_i, \hat{q}_i) + mse(p_i, \hat{p}_i). \quad (8)$$

Ablation studies in Sec. 4.3 demonstrate that Eq. (8) significantly outperforms Eq. (7). We conjecture that logit-level regularization provides a stronger and more direct training signal by enforcing consistency between raw logits, whereas probability-level consistency (after softmax) primarily aligns distributions, which does not necessarily ensure consistent decision boundaries.

# 4 EVALUATION

## 4.1 EXPERIMENTAL SETUP

**Datasets.** Following prior studies, we evaluate FedOpenMatch on three widely used benchmarks: CIFAR-10, CIFAR-100 (Krizhevsky et al., 2009), and SVHN (Netzer et al., 2011). To simulate data heterogeneity across clients, we sample client data partitions from a Dirichlet distribution $Dir(\alpha)$, where smaller $\alpha$ corresponds to higher heterogeneitsy. In our experiments, we set $\alpha \in \{0.1, 0.3\}$. For each dataset, a subset of classes is designated as seen classes, and a small fraction of them are selected as labeled data. For brevity, we denote the data configuration as $D@X@Y@Z$ where $D$ is the dataset name, $X$ is the number of seen classes, $Y$ is the number of labeled samples per class, and $Z$ indicates the partition setting.

**Baselines.** To validate the effectiveness of FedOpenMatch, we compare it against three categories of approaches: (i) *labeled data only*, which serves as the lower bound; (ii) *standard SSFL* methods; and (iii) *open-set SSFL* methods. For standard SSFL methods, we include representative works with publicly available implementations: SemiFL (Diao et al., 2022), FedLabel (Cho et al., 2023), and $FL^2$ (Lee et al., 2024). We exclude FedMatch since prior work (Diao et al., 2022) consistently reports its performance to be inferior to the labeled-only baseline. To the best of our knowledge, FedOpenMatch is the first attempt to tackle the open-set SSFL problem. As there are no direct baselines in this setting, we adapt recent OSSL methods originally designed for centralized training into the federated scenario, yielding federated counterparts of OpenMatch (Saito et al., 2021), SSB (Fan et al., 2023), IOMatch (Li et al., 2023b), and BDMatch (Hang & Zhang, 2024). Due to space limit, the implementation details are present in Appendix A.1.

**Evaluation Metrics.** Following prior work (Li et al., 2023b), we adopt two complementary metrics to assess performance: (i) **Closed-set accuracy** evaluates the ability of an OSSL algorithm to leverage open-set unlabeled data in order to improve classification performance on the seen classes only. (ii) **Open-set accuracy** measures performance in the realistic open-set environment, where both seen and unseen classes coexist. To enable evaluation, we treat all unseen classes as a single additional category, yielding a total of $K + 1$ classes. We report *Balanced Accuracy* (BA) as:

$$BA = \frac{1}{K+1} \sum_{k=1}^{K+1} \text{Recall}^k,$$

where $\text{Recall}^k$ denotes the recall for class $k$.

## 4.2 MAIN RESULTS

**Results on CIFAR-100.** Following prior works, we split CIFAR-100 according to super-classes. Experiments are conducted with either 80 or 50 seen classes, treating the remaining classes as un-

Table 2: Results of **closed-set accuracy** on **CIFAR-100** across eight different data settings.

| #Seen/Unseen classes | | | 80/20 | | 50/50 | |
|---|---|---|---|---|---|---|
| #Labels per class | | | 10 | 25 | 10 | 25 |
| | Partially | | 17.29±0.47 | 18.35±0.32 | 22.02±1.30 | 22.73±0.13 |
| Dir(0.3) | SemiFL | NeurIPS'22 | 29.64±0.47 | 38.81±1.83 | 43.45±1.20 | 53.82±1.56 |
| | FedLabel | ICCV'23 | 35.76±0.99 | 47.81±0.74 | 47.84±0.47 | 58.71±0.23 |
| | $FL^2$ | NeurIPS'24 | 35.08±0.01 | 45.23±0.01 | 46.10±0.79 | 53.95±0.84 |
| | OpenMatch | NeurIPS'21 | 22.25±2.63 | 34.58±1.59 | 32.93±1.05 | 28.68±16.9 |
| | SSB | ICCV'23 | 5.14±3.97 | 9.68±11.2 | 24.28±19.4 | 44.23±3.21 |
| | IOMatch | ICCV'23 | 37.59±1.29 | 47.50±1.09 | 48.35±1.49 | 56.44±1.23 |
| | BDMatch | ICML'24 | 34.05±0.98 | 42.19±1.33 | 46.57±2.27 | 37.45±25.9 |
| | FedOpenMatch | **Ours** | **39.45±1.04** | **52.96±0.67** | **49.94±1.72** | **60.19±0.97** |
| Dir(0.1) | SemiFL | NeurIPS'22 | 28.62±1.41 | 35.93±1.15 | 39.56±2.04 | 48.00±0.75 |
| | FedLabel | ICCV'23 | 32.88±0.82 | 41.40±1.24 | 44.13±1.76 | 50.37±0.73 |
| | $FL^2$ | NeurIPS'24 | 32.42±1.48 | 42.87±1.49 | 42.05±0.52 | 48.24±1.76 |
| | OpenMatch | NeurIPS'21 | 21.83±2.25 | 33.55±1.50 | 30.92±0.92 | 3.48±1.30 |
| | SSB | ICCV'23 | 8.93±2.93 | 9.30±13.9 | 36.14±1.65 | 30.55±19.6 |
| | IOMatch | ICCV'23 | 35.13±0.54 | 43.28±1.15 | 46.44±0.88 | 50.39±1.27 |
| | BDMatch | ICML'24 | 32.95±2.26 | 40.41±0.53 | 44.60±2.86 | 35.28±17.7 |
| | FedOpenMatch | **Ours** | **37.11±0.18** | **50.39±0.35** | **46.66±0.14** | **56.54±0.82** |

Table 3: Results of **open-set accuracy** on **CIFAR-100** across eight different data settings.

| #Seen/Unseen | | | 80/20 | | 50/50 | |
|---|---|---|---|---|---|---|
| #Labels per class | | | 10 | 25 | 10 | 25 |
| | Partially | | 16.08±0.31 | 17.71±0.67 | 21.29±1.27 | 17.76±0.44 |
| Dir(0.3) | OpenMatch | NeurIPS'21 | 2.11±0.77 | 1.74±0.33 | 1.97±0.01 | 1.99±0.04 |
| | SSB | ICCV'23 | 4.06±2.81 | 6.40±7.20 | 16.95±12.9 | 28.62±3.24 |
| | IOMatch | ICCV'23 | 28.66±0.85 | 37.72±2.06 | 42.86±0.83 | 49.45±1.92 |
| | BDMatch | ICML'24 | 23.19±1.86 | 29.38±2.53 | 39.15±3.39 | 31.14±25.4 |
| | FedOpenMatch | **Ours** | **38.97±1.03** | **50.40±0.42** | **46.29±0.87** | **59.01±0.95** |
| Dir(0.1) | OpenMatch | NeurIPS'21 | 2.41±0.66 | 2.46±0.78 | 2.12±0.27 | 2.36±0.38 |
| | SSB | ICCV'23 | 6.31±1.95 | 6.09±8.42 | 22.08±1.00 | 20.19±12.2 |
| | IOMatch | ICCV'23 | 27.20±1.37 | 33.02±0.33 | 41.10±0.32 | 43.39±0.80 |
| | BDMatch | ICML'24 | 23.20±1.30 | 29.51±0.99 | 33.46±2.76 | 30.54±15.3 |
| | FedOpenMatch | **Ours** | **36.65±2.43** | **47.35±2.98** | **43.91±0.65** | **54.60±1.53** |

seen. For each seen class, 10 or 25 samples are selected as the labeled set. Tab. 2 and Tab. 3 preset the results of closed-set accuracy and open-set accuracy, respectively. Since standard SSFL algorithms cannot detect outliers, open-set accuracy is only compared among OSSFL approaches. FedOpenMatch consistently outperforms existing methods on both metrics. For example, under the CIFAR-100@80@25@Dir(0.1) setting, closed-set and open-set accuracy are improved by up to 7.11% and 14.33%, respectively, demonstrating the effectiveness of our approach.

**Results on CIFAR-10 and SVHN.** For CIFAR-10, we follow prior studies (Li et al., 2023b) and select the animal classes as seen classes. For SVHN, classes 2 through 7 are treated as seen. For each seen class, 25 or 40 samples are randomly selected as the labeled set. Results of closed-set and open-set accuracy are presented in Tab. 4 and Tab. 5, respectively. The results highlight the improvements achieved by FedOpenMatch, particularly under highly heterogeneous settings (Dir(0.1)).

Table 4: Results of **closed-set accuracy** on **CIFAR10** and **SVHN**.

| Dataset-#Seen/Unseen classes | | CIFAR10-6/4 | | SVHN-6/4 | |
|---|---|---|---|---|---|
| #Labels per class | | 40 | 25 | 40 | 25 |
| Partially | | 42.47±0.73 | 38.35±1.11 | 70.29±4.83 | 51.55±3.88 |
| | SemiFL | NeurIPS'22 | 33.96±17.5 | 19.02±2.48 | 70.75±8.10 | 52.57±14.0 |
| | FedLabel | ICCV'23 | 70.21±1.09 | 64.09±2.08 | 83.75±0.15 | 79.63±1.54 |
| | $FL^2$ | NeurIPS'24 | 55.69±1.33 | 53.39±2.79 | 77.93±1.42 | 65.09±3.94 |
| Dir(0.3) | OpenMatch | NeurIPS'21 | 15.42±1.53 | 16.50±0.28 | 36.54±15.0 | 16.31±3.14 |
| | SSB | ICCV'23 | 16.67±0.00 | 16.67±0.00 | 12.41±0.00 | 16.19±3.28 |
| | IOMatch | ICCV'23 | 72.32±0.42 | **73.05±0.34** | 84.34±1.17 | **80.83±0.49** |
| | BDMatch | ICML'24 | 53.16±27.8 | 43.29±19.7 | 82.12±2.01 | 57.61±36.2 |
| | FedOpenMatch | **Ours** | **73.32±1.12** | 72.45±1.26 | **84.86±1.57** | 80.03±1.76 |
| | SemiFL | NeurIPS'22 | 28.50±16.7 | 17.26±0.83 | 70.03±4.41 | 47.05±4.80 |
| | FedLabel | ICCV'23 | 59.36±0.59 | 54.22±1.93 | 77.67±2.64 | 69.30±3.37 |
| | $FL^2$ | NeurIPS'24 | 50.82±2.07 | 45.29±3.24 | 72.73±2.53 | 65.43±2.59 |
| Dir(0.1) | OpenMatch | NeurIPS'21 | 16.68±0.16 | 16.63±0.04 | 63.66±12.3 | 13.71±2.02 |
| | SSB | ICCV'23 | 16.67±0.00 | 16.67±0.00 | 12.41±0.00 | 14.30±3.28 |
| | IOMatch | ICCV'23 | 64.79±1.55 | 59.60±0.39 | 79.48±0.43 | 76.51±1.35 |
| | BDMatch | ICML'24 | 36.87±21.7 | 51.73±3.22 | 80.92±1.39 | 58.33±28.3 |
| | FedOpenMatch | **Ours** | **65.19±2.83** | **60.89±0.71** | **82.63±1.50** | **78.36±0.74** |

Table 5: Results of **open-set accuracy** on **CIFAR10** and **SVHN**.

| Dataset-#Seen/Unseen | | CIFAR10-6/4 | | SVHN-6/4 | |
|---|---|---|---|---|---|
| #Labels per class | | 40 | 25 | 40 | 25 |
| Partially | | 36.02±0.66 | 32.32±0.75 | 60.44±4.59 | 43.93±1.71 |
| | OpenMatch | NeurIPS'21 | 14.28±0.00 | 14.28±0.00 | 14.29±0.00 | 14.28±0.00 |
| | SSB | ICCV'23 | 14.39±0.18 | 14.42±0.24 | 14.29±0.00 | 14.30±0.03 |
| Dir(0.3) | IOMatch | ICCV'23 | 61.65±0.48 | 61.82±0.42 | 71.78±1.30 | **68.52±0.88** |
| | BDMatch | ICML'24 | 42.62±24.5 | 24.59±17.8 | 69.36±1.48 | 48.32±29.5 |
| | FedOpenMatch | **Ours** | **62.26±0.51** | **61.98±2.18** | **71.87±0.83** | 67.75±2.57 |
| | OpenMatch | NeurIPS'21 | 14.28±0.00 | 14.29±0.00 | 53.59±15.2 | 14.29±0.00 |
| | SSB | ICCV'23 | 14.30±0.02 | 14.33±0.07 | 15.65±2.14 | 14.48±0.34 |
| Dir(0.1) | IOMatch | ICCV'23 | 54.94±1.37 | 50.71±1.78 | 68.13±0.21 | 64.41±1.15 |
| | BDMatch | ICML'24 | 25.86±20.1 | 24.65±17.9 | 67.49±1.18 | 46.47±27.9 |
| | FedOpenMatch | **Ours** | **55.27±2.21** | **51.45±0.45** | **70.57±0.93** | **66.70±1.02** |

**Comparison under complex data settings.** To better simulate real-world data complexity, we construct a variant of CIFAR-100, denoted as CIFAR100-C, by applying diverse corruption types following the protocol of (Hendrycks & Dietterich, 2019)[1]. In this dataset, 70% of the samples are corrupted to induce feature shift, while the remaining 30% are kept clean. Each client then samples its local data according to a Dirichlet distribution (consistent with our main experiments) to additionally introduce label shift. This setup reflects realistic scenarios where both feature shift and label shift coexist. Under these challenging conditions, FedOpenMatch still achieves the best performance by a clear margin as shown in Fig. 13 in Appendix A.2, demonstrating strong robustness to complex and heterogeneous data distributions.

**Comparison under scenarios with high unknown-class prevalence.** To further evaluate FedOpen-Match under scenarios where unseen prevalence is very high, we conduct experiments on CIFAR-

---

[1]We follow the official instructions at https://github.com/hendrycks/robustness.

Table 6: Ablation studies isolating each component.

| CIFAR100@80@25@Dir(0.1) | Base | Base+GS | Base + GS + LA | Base + GS + LA + LCR |
|---|---|---|---|---|
| Open-set Accuracy | 34.81 | 39.51 | 43.65 | 48.36 |
| Closed-set Accuracy | 45.86 | 46.14 | 45.81 | 50.38 |

100 dataset with only 20 seen classes and the remaining 80 classes are unknown. Figure 14 in Appendix A.2 shows that FedOpenMatch significantly outperforms baselines, demonstrating its effectiveness when unknown classes dominate the data distribution.

**Summary.** The results above show that directly adapting OSSL algorithms to the federated setting leads to unstable performance. For instance, SSB and OpenMatch often fail to surpass the lower-bound baseline, despite sharing the same backbone architecture as our method. To investigate the cause, we visualize the utilization rate and pseudo-label accuracy in Figs. 10 and 11 in Appendix A.2. As illustrated, OpenMatch maintains a persistently low utilization rate of unlabeled data throughout training. This is mainly due to its "detect-and-filter" strategy, where the imbalanced OVA classifier mistakenly identifies most samples as outliers, leaving a large portion of informative data underutilized. This observation highlights the necessity of logit adjustment. In contrast, SSB achieves higher utilization by incorporating all samples with confident inlier predictions, regardless of whether they are inliers or outliers. However, its pseudo-label accuracy remains low, which ultimately constrains overall performance. Since clients in OSSFL lack reliable supervision, the models are highly susceptible to incorrect supervision, leading to progressively worse pseudo-labels.

FedOpenMatch addresses these challenges by generating pseudo-labels with the global model and keeping them fixed during local training, thereby avoiding degradation of pseudo-label quality caused by unstable updates. Coupled with the proposed components, the data utilization rate steadily increases while maintaining relatively high pseudo-label accuracy. As a result, FedOpenMatch achieves consistent state-of-the-art performance across nearly all settings, demonstrating both its robustness and superiority, particularly under challenging data configurations.

## 4.3 ABLATION STUDIES

In this section, we evaluate the effectiveness of key components in FedOpenMatch, and examine its sensitivity to hyper-parameters. Due to space limit, we put the detailed results in Appendix A.2.

**Ablation studies isolating each component.** We conduct experiments to demonstrate the effectiveness and necessity of each component, as summarized in Tab. 6. The results show the effectiveness and necessity of each module, confirming that each component contributes meaningfully to the overall performance of FedOpenMatch.

**Effect of gradient stop (GS).** We conduct experiments under CIFAR100@50@10@– configurations to investigate the effect of GS. As shown in Fig. 3 and Fig. 4 in Appendix A.2, with the adoption of GS, the performance is consistently improved, with the open-set accuracy gap reaching up to 10.68%. Meanwhile, we observe that the gradient similarity between the inlier classifier and OVA classifier branches is highly unstable. When their optimization directions happen to align, the two branches temporarily reinforce each other (e.g., in the early training stage without GS, the accuracy can exceed that with GS). However, as the gradient similarity drops, the branches interfere with each other and overall performance degrades. In other settings (e.g., Fig. 5 in Appendix A.2), where the similarity remains low throughout training, the performance without GS is consistently worse than with GS. Overall, GS is important for achieving stable and superior performance.

**Effect of logit adjustment.** To mitigate the imbalance between inliers and outliers in OVA training (Sec. 3.3), we introduce logit adjustment. We evaluate its effect on CIFAR-100@80@25@Dir(0.1) under different adjustment weights $\omega$. As shown in Fig. 7 and in Fig. 8 in Appendix A.2, open-set accuracy drops significantly (18.19%) when $\omega = 0$, as the OVA classifier becomes biased toward the majority classes (unseen classes). Performance peaks when $\omega = 1$, validating the effectiveness of logit adjustment in correcting class imbalance.

**Effect of logit consistency regularization.** We propose a weak-strong logit consistency regularization to better exploit unlabeled samples (Sec. 3.4). We study the effect of the consistency weight $\lambda$ on CIFAR-100, with results in Fig. 7 in Appendix A.2. Both closed-set and open-set accuracy are sensitive to $\lambda$. When $\lambda = 0$, closed-set and open-set accuracy drop by 4.82% and 5.61%, respectively, indicating the necessity of logit consistency. The best closed-set accuracy is achieved at $\lambda = 2$ (improving 0.9% over $\lambda = 1$), but with a 4.7% drop in open-set accuracy. Overall, $\lambda = 1$ provides the best trade-off between closed-set and open-set performance. We further compare our proposed logit consistency loss (Eq. (8)) with the probability consistency loss (Eq. (7)). As shown in Fig. 7 in Appendix A.2, our LCR loss significantly boosts both closed-set and open-set accuracy, for example, by 4.83% and 3.68% under data configuration CIFAR100@80@25@Dir(0.1), demonstrating its superior ability to transfer richer information.

**Influence of local epochs.** To understand the influence of client local training epochs, we conduct additional experiments on CIFAR-100 dataset. As shown in **??** in Appendix A.2, using too few local epochs leads to underfitting and slow convergence, requiring many more communication rounds to achieve reasonable performance. In contrast, using too many local epochs can improve accuracy but introduces the risk of unstable or divergent training due to excessive local drift. The conclusion is consistent with prior SSFL work (Diao et al., 2022).

**Influence of OVA threshold.** To understand the influence of OVA threshold, we conduct experiments on CIFAR-100 with varying OVA threshold. The results are presented in Fig. 12 in Appendix A.2, which suggests that closed-set accuracy remains stable across a wide range of OVA thresholds, indicating that this component has limited impact on inlier classification. In contrast, open-set accuracy is more sensitive to the threshold value. A low threshold admits more samples during training but also introduces substantial noise, leading to unstable optimization and degraded open-set performance. Therefore, using a higher threshold is generally preferable, as it filters out unreliable samples and stabilizes training.

**Comparison with different backbones.** To further assess the effectiveness and robustness of FedOpenMatch across model architectures, we conduct additional experiments using WideResNet-28×2, a widely adopted backbone in the SSL and SSFL literature. As shown in Tab. 7 in Appendix A.2, FedOpenMatch consistently achieves the best performance, demonstrating that its components do not rely on any backbone-specific assumptions.

## 5 CONCLUSION

In this work, we formulate the practical yet underexplored problem of open-set semi-supervised federated learning, where the distributed unlabeled dataset contains samples from classes unseen in the labeled data. To address this challenge, we propose the first OSSFL framework, FedOpenMatch, which effectively leverages distributed open-set unlabeled data to improve model performance under both closed-set and open-set scenarios. We conduct extensive experiments on widely used benchmark datasets, demonstrating that, despite its simplicity, FedOpenMatch outperforms existing approaches by a clear margin, particularly under complex data configurations. We also provide detailed analyses to validate the contribution of each proposed component.

## 6 LIMITATION AND FUTURE WORK

While FedOpenMatch demonstrates significant empirical gains, a rigorous theoretical understanding of its underlying mechanisms remains incomplete – an open challenge shared by many SSL approaches. We leave a more formal theoretical analysis to future work. In addition, several promising extensions remain. For example, the inlier and OVA classifiers may complement each other when their optimization directions are aligned, suggesting the potential of adaptively controlling the gradient stop mechanism or explicitly promoting consistency between their optimization processes.

### ACKNOWLEDGEMENTS

The computations in this research were performed using the CFFF platform of Fudan University. Jihong Guan was supported by National Natural Science Foundation of China (NSFC) under grant No. 62372326.

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

Table 7: Performance comparison of FedOpenMatch with state-of-the-art baselines using a different backbone (WideResNet-28×2). The results show that FedOpenMatch consistently achieves the best performance with a clear margin, demonstrating its robustness to different backbone architectures.

| CIFAR-100 | 50@25@Dir(0.1) Open-set Acc | 50@25@Dir(0.1) Closed-set Acc | 80@25@Dir(0.1) Open-set Acc | 80@25@Dir(0.1) Closed-set Acc |
|---|---|---|---|---|
| FedLabel | None | 47.12 | None | 37.05 |
| IOMatch | 35.71 | 44.58 | 24.65 | 37.12 |
| FedOpenMatch | 50.16 | 52.14 | 42.69 | 43.23 |

## A  Experimental Details

### A.1  Implementation

In our experiments, we employ ResNet-18 (He et al., 2016) as the backbone for all datasets and methods, with the classification head implemented as described in the original papers. The number of clients is set to 50 for CIFAR100@50@-@- and 100 for the other settings. In each communication round, 10 clients are randomly selected for local training. Following prior work, we assign equal weight to each local model for aggregation.

We train the models for 600 communication rounds and adopt SGD with a momentum of 0.9 and weight decay of 0.0005 as the optimizer. The learning rate is initialized at 0.03 and decayed using a cosine schedule. The number of local training epochs for clients and server are set to $E_c = 5$ and $E_s = 5$, respectively. For all methods, the client batch size is 32, and the server batch size is 10 or 250 depending on whether the number of labeled samples is fewer or greater than 1000. Following prior work, we employ confidence thresholds $\tau_{in} = 0.95$ for inlier pseudo-labels, and $\tau_{pos} = 0.99$, $\tau_{neg} = 0.01$ for OVA pseudo-labels. The weight for logit adjustment is set to $\omega = 1$ and the weight of LCR (Eq. (8)) is set to $\lambda = 1$ across all settings.

**Fair Comparison.** To ensure a fair comparison, we implement the federated versions of the baseline OSSL algorithms following their official implementation and use the hyperparameters recommended in their respective papers. The data distribution is consistent across all methods. We conduct *three independent runs* on all datasets using random seeds (0, 10, and 100). After each communication round, we evaluate the global model on the original test set and report the mean and standard error of closed-set accuracy and open-set accuracy.

All experiments are conducted on an Ubuntu 16.04 server using PyTorch 1.8.0. The **source code** and detailed library dependencies are provided in Supplementary Material.

### A.2  Detailed experimental results.

Due to the space limitation, we present the detailed experimental results here.

**Computation/communication overhead.** It is worth-noting that OVA classifier does not introduce much additional communication and computation overhead as it only consists of a two-layer projection head and a linear classification head. The number of computation and communication overhead are negligible relative to the backbone.

## B  The use of large language models

During the preparation of this manuscript, we used ChatGPT to assist with writing by providing the prompt: "I am preparing a paper for submission to an international conference and would like your help to check for any grammatical issues and refine the wording or sentence structure where necessary to ensure conciseness and precision." We applied its suggestions paragraph by paragraph, and all outputs were edited by us to ensure correctness.

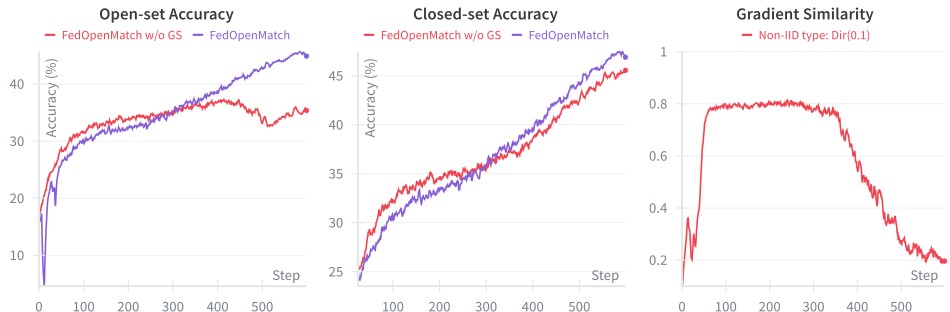

Figure 3: Performance comparison of FedOpenMatch with and without the Gradient Stop (GS) strategy on CIFAR-100@50@10@Dir(0.1), along with the gradient similarity between the inlier classifier and OVA classifier branches. In the early training stage, the performance without GS surpasses that with GS, suggesting that the OVA classifier and inlier classifier can reinforce each other when their update directions are aligned. However, as training progresses, the gradients begin to diverge, and the resulting interference between the two classifiers leads to a performance drop.

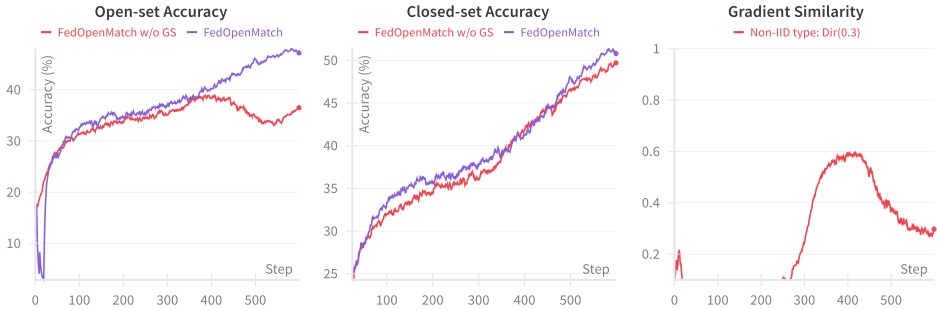

Figure 4: Performance comparison of FedOpenMatch with and without the Gradient Stop (GS) strategy on CIFAR-100@50@10@Dir(0.3), along with the gradient similarity between the inlier classifier and OVA classifier branches. Since the update directions diverge for most of the training process, the performance with GS surpasses that without GS, as expected.

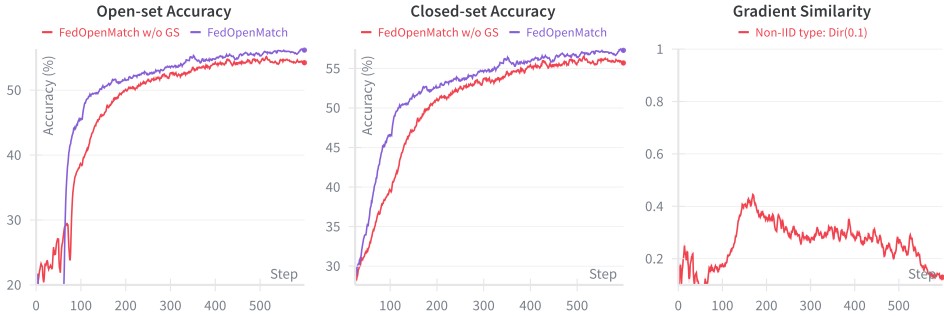

Figure 5: Performance comparison of FedOpenMatch with and without the Gradient Stop (GS) strategy on CIFAR-100@50@25@Dir(0.1), along with the gradient similarity between the inlier classifier and OVA classifier branches. The gradient similarity remains low throughout the training, thus the performance with GS consistently surpasses that without GS.

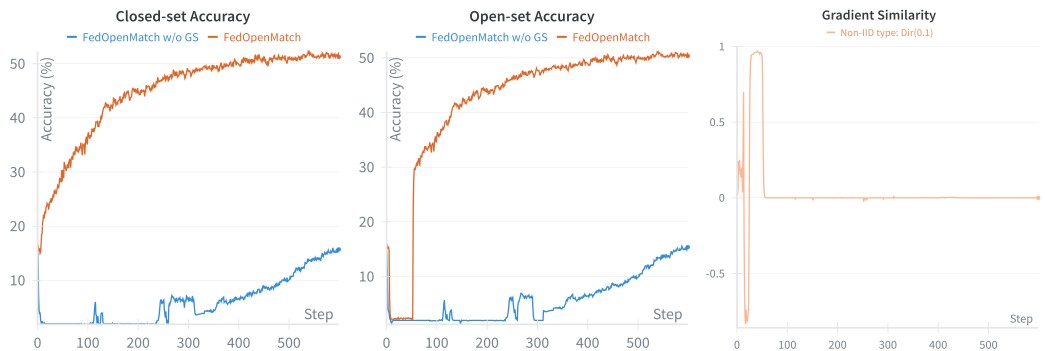

Figure 6: Performance comparison of FedOpenMatch with and without the Gradient Stop (GS) strategy on CIFAR-100@50@25@Dir(0.1) **with different backbone (WideResNet-28x2)**, along with the gradient similarity between the inlier classifier and OVA classifier branches. The gradient similarity remains extremely low during the whole training process, which proves the clear performance gap between models with and without GS.

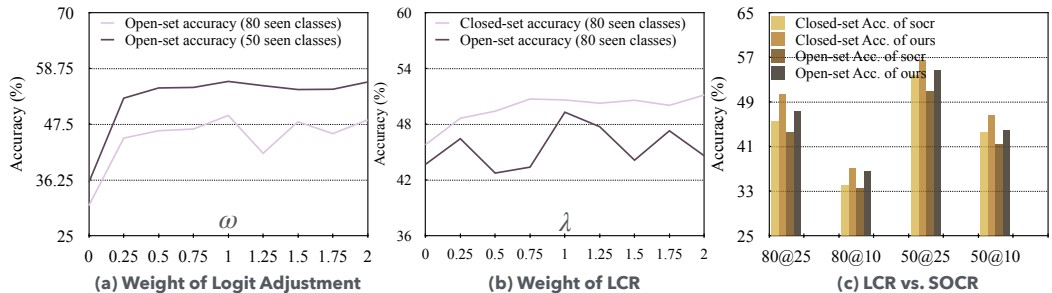

Figure 7: Ablation studies. For simplicity, we use X@Y to represents data setting with X seen classes and Y labeled samples per class. (**a**): FedOpenMatch with different $\omega$ under CIFAR-100@-@25@Dir(0.1) setting. Without logit adjustment ($\omega = 0$), the open-set accuracy degrades significantly. (**b**): FedOpenMatch with different $\lambda$ under CIFAR-100@80@25@Dir(0.1) setting. The performance is improved with LCR loss. (**c**): Comparison of existing **SOCR** loss and our **LCR** loss under CIFAR100@-@-@Dir(0.1) setting. Our LCR outperforms existing SOCR across various settings.

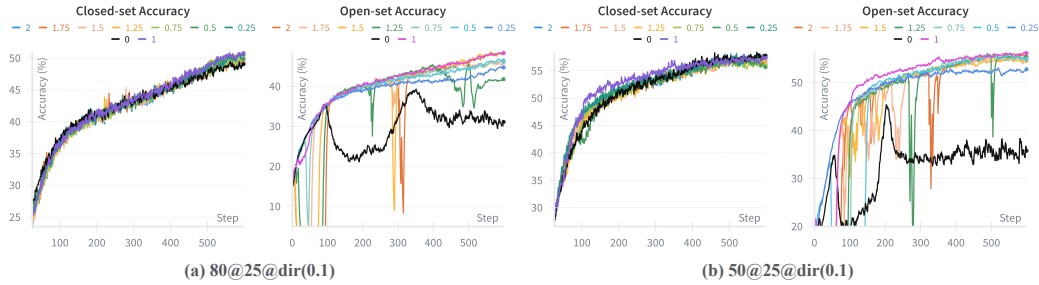

Figure 8: Learning curves of training with various strength of logit adjustment (Eq. (6)) on CIFAR-100 dataset. The results present a clear accuracy margin between those with and without logit adjustment.

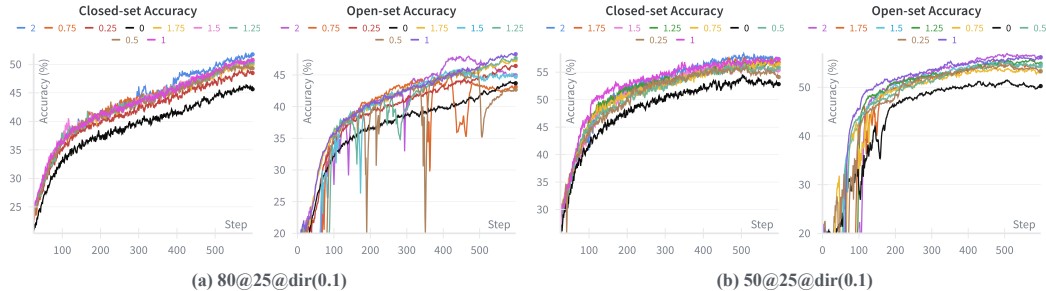

Figure 9: Learning curves of training with various weights of logit consistency regularization loss (Eq. (8)) on CIFAR-100 dataset. The performance gets clear improvement with logit consistency regularization loss.

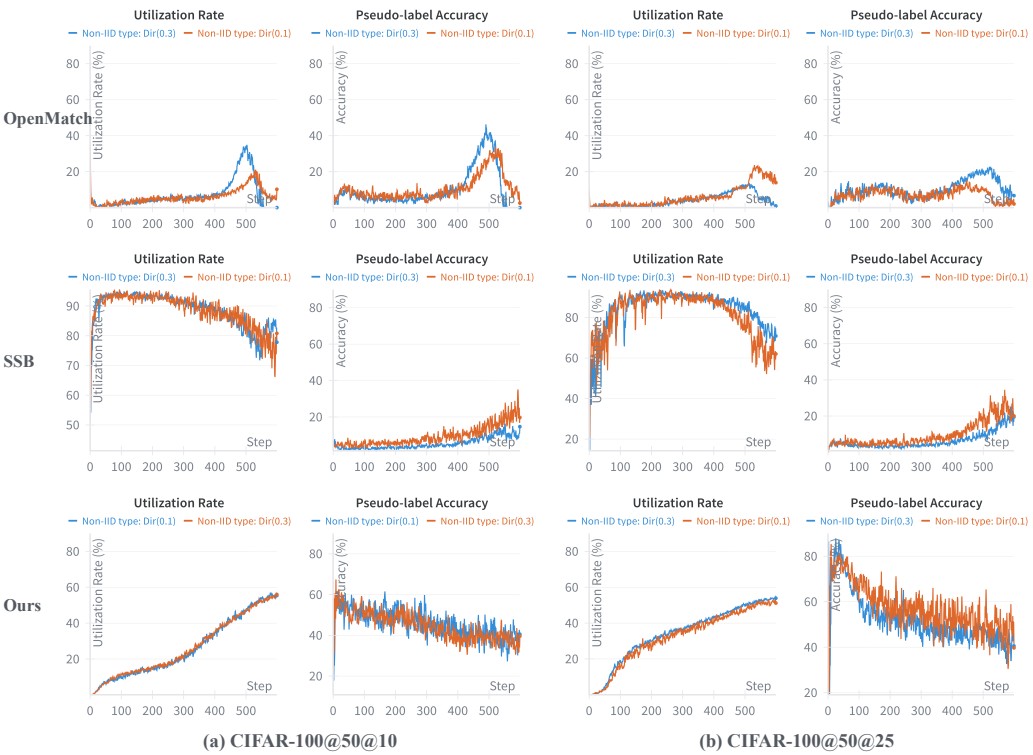

Figure 10: Comparison of **utilization rate** and **pseudo-label accuracy** of OpenMatch (Saito et al., 2021), SSB (Fan et al., 2023) and FedOpenMatch (ours), which share the same architecture. Open-Match suffers from both a low data utilization rate and poor pseudo-label quality, as most samples are mistakenly identified as outliers by the imbalanced OVA classifier, thereby limiting its performance. SSB relaxes the pseudo-label filtering strategy to increase data utilization, but this introduces a large number of incorrect pseudo-labels. In contrast, FedOpenMatch addresses both issues simultaneously, achieving higher data utilization and more reliable pseudo-label accuracy.

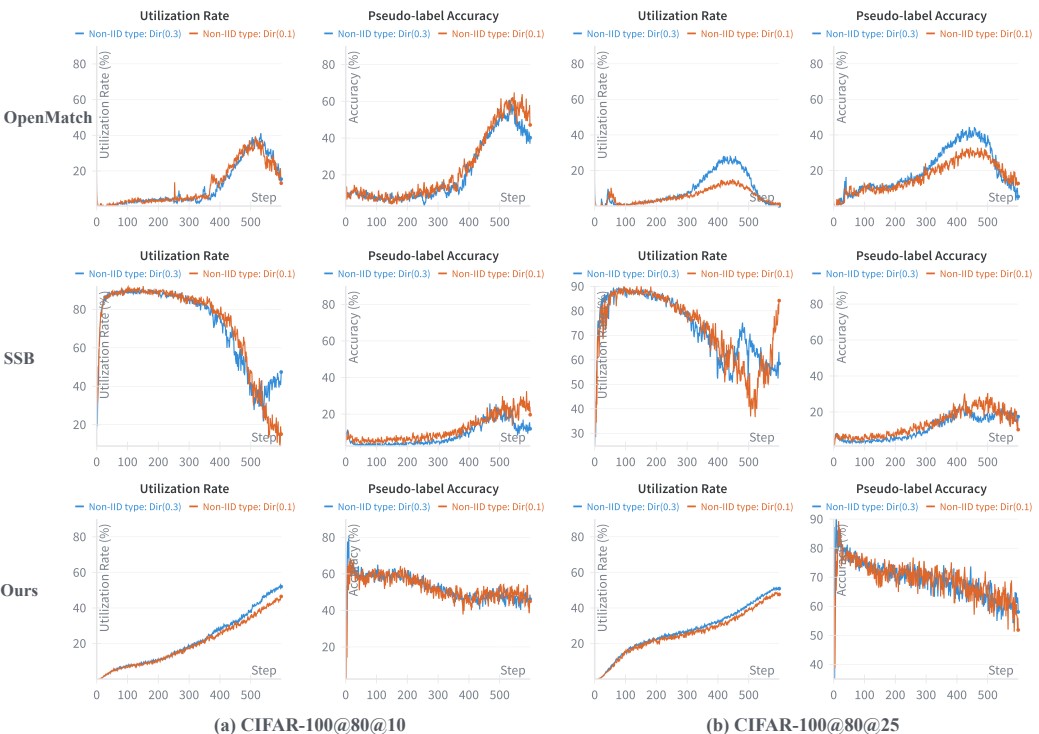

Figure 11: Comparison of **utilization rate** and **pseudo-label accuracy** of OpenMatch (Saito et al., 2021), SSB (Fan et al., 2023) and FedOpenMatch (ours), which share the same architecture. OpenMatch suffers from both a low data utilization rate and poor pseudo-label quality, as most samples are mistakenly identified as outliers by the imbalanced OVA classifier, thereby limiting its performance. SSB relaxes the pseudo-label filtering strategy to increase data utilization, but this introduces a large number of incorrect pseudo-labels. In contrast, FedOpenMatch addresses both issues simultaneously, achieving higher data utilization and more reliable pseudo-label accuracy.

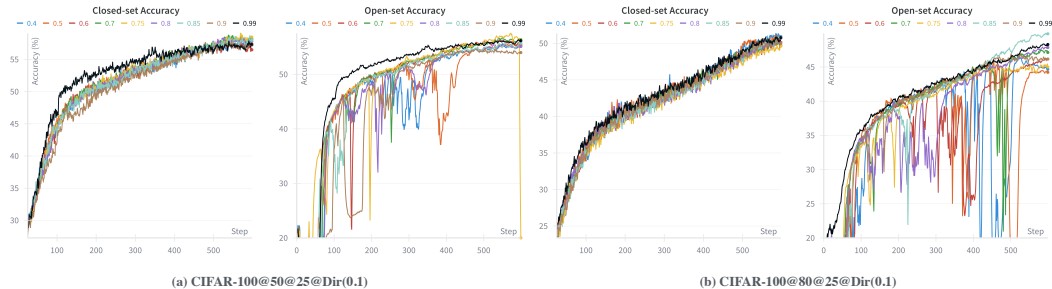

Figure 12: Performance comparison of FedOpenMatch under different OVA thresholds on the CIFAR-100 dataset. The results show that closed-set accuracy remains stable across a wide range of OVA thresholds, indicating that this component has limited impact on inlier classification. In contrast, open-set accuracy is more sensitive to the threshold value. A low threshold admits more samples during training but also introduces substantial noise, leading to unstable optimization and degraded open-set performance. Therefore, using a higher threshold is generally preferable, as it filters out unreliable samples and stabilizes training.

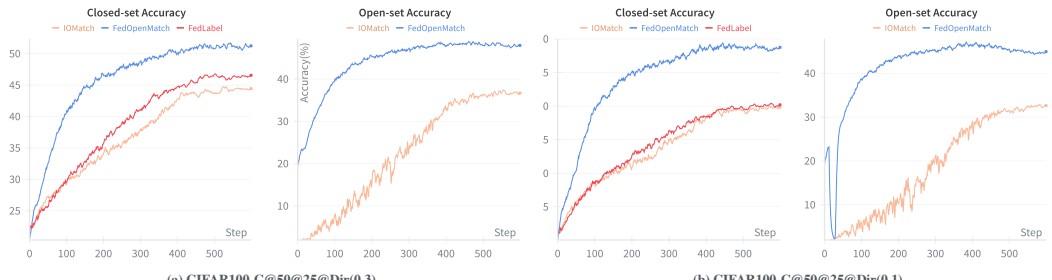

Figure 13: Performance comparison under complex settings where label shift and feature shift coexist. We further evaluate FedOpenMatch in a more realistic scenario that introduces both label shift and feature shift simultaneously. We compare against FedLabel and IOMatch, the two strongest-performing baselines. Across all metrics, FedOpenMatch maintains a clear performance margin over both methods, demonstrating its robustness and superiority under challenging, real-world data heterogeneity.

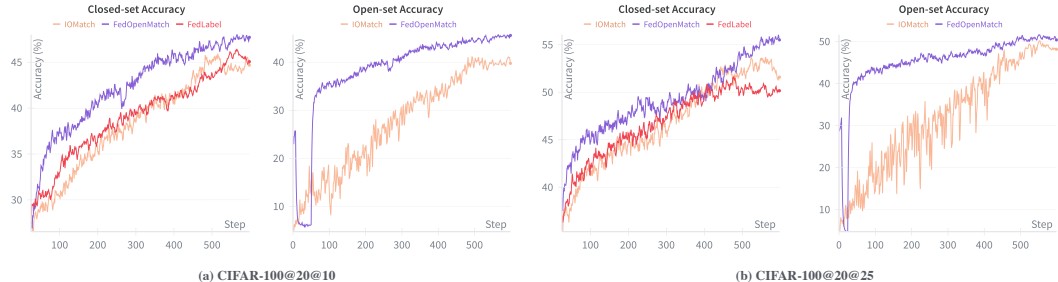

Figure 14: Performance comparison under extreme scenarios with high unknown-class prevalence. We further evaluate FedOpenMatch on CIFAR-100 with only 20 seen classes and 80 unseen classes, creating an extreme setting where unknown samples dominate the data. Even in this highly challenging scenario, FedOpenMatch consistently achieves the best performance among all methods, demonstrating strong robustness and clear superiority under extreme open-set conditions.

