# OpenReview forum: "FedOpenMatch: Towards Semi-Supervised Federated Learning in Open-Set Environments"
_ICLR.cc/2026/Conference — ICLR 2026 Poster_

### Official Review · Reviewer_abnT · 2025-10-19

**Soundness:** 3
**Presentation:** 3
**Contribution:** 2
**Rating:** 6
**Confidence:** 2

**Summary:**

This paper proposes a method called FedOpenMatch for open-set federated semi-supervised learning. By "open-set", the authors refer to the setting where outlier classes exist in unlabeled datasets, and the model should be able to classify inlier classes right, while detecting outlier classes. To address the problem, FedOpenMatch leverages a two-head structure, with a feature extractor, an inlier classifier and an OVA outlier classifier. The authors introduce several techniques to enhance the performance of FedOpenMatch, such as stop-gradients for the OVA branch, logit consistency instead of probability consistency, and logit adjustment to handle the imbalance issue of OVA classifiers. Experiments are done on three datasets and against multiple baselines, where FedOpenMatch outperforms a variety of baselines, including open-set and closed-set ones.

**Strengths:**

1. This paper tackles the problem of open-set federated semi-supervised learning, which is of practical value. Being able to detect outlier classes is an important property especially in FL, where the collected data may contain a large amount of noise.
2. This paper is overall well-written and easy to follow. The organization of this paper is reasonable, and the technical solutions and intuitions are clearly stated together, making it easy to understand.
3. Experiments of FedOpenMatch is extensive. The authors compare against lots of baselines, including SSFL algorithms and adapted Open-set SSL methods (to the FL setting) with multiple baselines, number of seen classes, etc. Moreover, ablation studies are sufficient and demonstrate the impact of each individual design technique. Some ablation studies even analyze why the proposed techniques help (e.g. analyzing gradient similarity to understand the impact of gradient stop). This makes the paper well-justified.

**Weaknesses:**

1. One major weakness of this paper is that it is built upon multiple existing ideas. For example, the logit adjustment method is directly taken from Menon et al. 2021, and the weak-strong consistency is modified from existing methods. Overall, this slightly weakens the amount of novel insights of this paper, despite still being a solid paper with new problems tackled.
2. I am interested in how FedOpenMatch is sensitive to the number of local iterations/epochs. As local clients only have unlabeled data, training on solely unlabeled data for too long may lead to diverging model updates. Therefore, from my understanding, the number of local iterations/epochs is an important parameter to determine in federated SSL. The authors did not provide such analyses though.
3. Gradient similarity fail to completely explain the accuracy curve. While in Figure 3, the relations between gradient similarity and open-set accuracy is significant, such relations are not so apparent in Figures 4 and 5. For example, in Figure 4, for the early ~250 steps, gradient similarity is below 0.2, yet the accuracies are close. This makes me wonder whether there are other factors that may impact the accuracy, e.g. gradient magnitude.
4. Minor points.
    - It seems that in Eqn. 6, the sign of $\omega\log \pi$ is reversed. I checked the original paper and found that the sign before the term should be - instead of +.

**Questions:**

One minor question about FedOpenMatch is when a sample will be categorized as outlier, e.g. when all OVA classifiers report it as an outlier? I am not highly familiar with open-set semi-supervised learning so some additional backgrounds may help.

---

> ### Author Response · Authors · 2025-11-20
>
> Thank you very much for your insightful and constructive comments. Our responses are presented as follows:
>
> **(W1) Novelty.**  We acknowledge that logit adjustment and consistency regularization originate from prior works. However, our contribution is not merely reusing these ideas, but identifying the unique failure modes of OSSFL and designing principled adaptations that make them effective in our setting. Each component in FedOpenMatch is motivated by a concrete OSSFL challenge:
> 	1.	Gradient-stop is introduced after diagnosing severe gradient conflict between the inlier and OVA classifiers under federated non-IID updates.
> 	2.	Logit adjustment, though originally proposed elsewhere, is adopted here to correct the strong prediction bias intrinsic to OVA classifiers, which otherwise severely limits useful data utilization.
> 	3.	Logit-level consistency is a subtle but crucial modification—while prior SSL/OSSL works typically use probability-level consistency, we found that logit-level consistency yields significantly better stability and performance in the OSSFL setting (Fig.6(c)).
> **Thus, although the components draw inspiration from existing techniques, their integration and adaptation are novel and directly driven by OSSFL-specific challenges. The resulting method is simple yet effective, and we believe it provides valuable insights for future research in this emerging area.**
>
> **(W2) Sensitivity to local epochs.** We agree that the number of local iterations is a crucial factor in FL, as also analyzed in SemiFL. Small local epochs may cause underfitting and require many communication rounds, while large local epochs may lead to overfitting, severe update divergence, and instability under non-IID data. To ensure fairness and stability, we therefore follow the recommended local epochs from prior SSFL works and apply the same configuration to all algorithms in our experiments.
> To further demonstrates the influence, **we conduct additional experiments with 1/5/10 local epochs. The results in Fig. 11 (revised version) lead to consistent conclusion with prior work.**
>
> **(W3) Gradient similarity.**
> We thank the reviewer for this insightful observation.
> We agree that gradient similarity alone does not fully explain the accuracy dynamics. Our goal is to show that it reveals the underlying optimization conflict between the inlier and OVA classifiers—the very issue that Gradient Stop is designed to mitigate.
> In the early training stage, the backbone is not yet well trained, so accuracy differences remain small even when similarity is low or high. This is consistent with Fig.3, where similarity is high early on but the accuracy gap has not emerged yet.
> Although gradient similarity is not a complete predictor of accuracy at every step, it successfully highlights the core problem: the OVA branch often produces gradients that diverge from the inlier branch, which leads to instability and degraded performance without Gradient Stop.
>
> **(W4)** Thank you for carefully checking the equation. The confusion arises from the fact that there are two forms of logit adjustment in the original paper:
> **Post-hoc adjustment (applied at inference) uses $– log\pi_y$** , which is what the reviewer refers to. **Training-time logit adjustment, which we adopt, uses $+ log\pi_y$**, as shown in Eq. (10) of the original paper.
> Our Eqn. 6 follows the training-time formulation, not the post-hoc version. Therefore, the sign in our equation is correct.

---

> > ### Comment · Reviewer_abnT · 2025-11-22
> > **Thank you for your response.**
> >
> > I have read the author's response. The response is extensive and covered most of my points in the review.
> >
> > I agree this is a solid paper with useful insights in practice (and the rebuttal reinforced my opinion on that). However, as this paper is still more like a combination of existing techniques, I would not raise my rating.
> >
> > Thanks and best of luck.

---

> > > ### Author Response · Authors · 2025-11-24
> > >
> > > Dear Reviewer abnT:
> > >
> > > Thank you very much for your timely response and for your supportive comments. We truly appreciate your recognition of the solidity of our work.
> > >
> > > Regarding the novelty, we would like to emphasize that our paper proposes the first Open-set SSFL framework. The Gradient Stop mechanism is newly introduced in both centralized and federated open-set semi-supervised learning settings and is supported by extensive evidence.  In addition, the logit-level consistency regularization represents an impactful modification to the widely used probability-level consistency in existing SSL/OSSL literature. We believe that FedOpenMatch would be beneficial to future researches in this area.
> > >
> > > Thank you again for your constructive feedback and support.
> > > We sincerely wish you all the best.
> > >
> > > The Authors

---

### Official Review · Reviewer_5NdG · 2025-10-29

**Soundness:** 3
**Presentation:** 2
**Contribution:** 2
**Rating:** 4
**Confidence:** 4

**Summary:**

The paper introduces the open-set semi-supervised federated learning (OSSFL) problem, where clients possess unlabeled data potentially containing unseen classes, and the server maintains a small labeled set. To address this, the authors propose FedOpenMatch, a framework that jointly trains an inlier classifier and a one-vs-all (OVA) outlier detector. The method incorporates gradient stop, logit adjustment, and logit consistency regularization to improve stability and open-set discrimination under federated conditions.

**Strengths:**

The paper presents a clear problem setting that distinguishes SSFL, OSSL, FOSR, and the OSSFL. By adding the GS, logit adjustment and LCR to a standard architecture, experimental results show FedOpenMatch improves both closed-set and open-set balanced accuracy over federated adaptations of OSSL baselines and standard SSFL methods.

**Weaknesses:**

While the paper positions as a new OSSFL formulation, its novelty relative to prior open-world SSFL work is limited and largely rooted in scenario framing rather than a new learning principle. The authors should better justify why the label-at-server configuration represents the dominant or more practical case, and clarify how insights generalize across settings. As it stands, the novelty appears to hinge more on scenario framing than on a fundamentally new learning principle.

Additionally, centralized OSSL algorithms appear to be adapted to FL. Their relatively weak and unstable performance raises concern that implementations or hyperparameters under FL are suboptimal. More evidence could be helpful to show these are strong federated instantiations rather than strawmen.

Further, the evaluation setup is narrow, using only ResNet-18, limited Dirichlet heterogeneity, and no covariate or mixed-shift scenarios. Key design details, such as pseudo-label staleness under non-IID drift, are under-analyzed. Finally, grouping all unseen classes into a single "unknown" category obscures variability.

**Questions:**

N/A

---

> ### Author Response · Authors · 2025-11-20
>
> Thank you very much for your comments. We present our responses to your comments in detail as follows:
>
> **(W1) Novelty.**  We respond to your comment from three aspects:
>
> 1.**Our work is different from existing FedoSSL. Existing open-world SSFL and our open-set SSFL are distinct problems**. The former focuses on learning new discovered classes, while the latter concentrates on reliable rejection of unseen classes. The existing **FedoSSL further assumes that every client has labeled data—an unrealistic requirement**. **Our OSSFL formulation adopts label-at-server, with clients holding only unlabeled, heterogeneous data**, leading to fundamentally different and more challenging learning dynamics.
>
> 2.**Label-at-server is more practical and challenging**. In practical FL, clients usually lack labeling expertise and resources, whereas servers have more resources to curate a small clean labeled set. Our setup uses **0.2% labeled data**, while the existing FedoSSL assumes **50% labeled client data**, which is impractical. This makes our problem harder as clients rely entirely on pseudo-labels.
>
> 3.**Our work’s novelty is beyond framing.** Our contributions stem from a deep analysis on OSSFL-specific failure modes — severely imbalanced OVA training and conflicting gradients, and correspondingly introducing targeted designs (gradient-stop, logit adjustment, logit-level consistency) that enable stable and effective OSSFL.
>
> **(W2) Fair Comparison.**
>
> We appreciate the reviewer’s concern and agree that strong, fair baselines are essential. Our goal is to show how existing OSSL methods behave when ported to FL. To ensure fairness, we implemented all centralized OSSL algorithms strictly from their official source code, used the method-specific hyperparameters exactly as recommended in the original papers, and adopted FL hyperparameters (optimizer, learning rate schedule, local epochs, augmentations, etc.) directly from prior SSFL works such as SemiFL and FL². We did not tune hyperparameters in favor of our method. **Full hyperparameter sweeps are infeasible**—each OSSFL experiment takes more than a day and must be repeated across every data settings—**so following established SSFL conventions ensures a fair and reproducible comparison.**
>
> Importantly, **the weak or unstable performance is intrinsic to centralized OSSL when placed in the FL setting.** Without correct label supervision, client-side local updates rapidly cause pseudo-label drift: degraded pseudo-labels lead to degraded models, which in turn produce even noisier pseudo-labels. Moreover, prior OSSL methods either fail to correct the OVA classifier’s severe imbalance—resulting in many discarded samples—or fail to maintain high pseudo-label accuracy, as evidenced in Fig.9 and Fig.10. These limitations naturally manifest under non-IID federated training, explaining their instability.

---

> ### Author Response · Authors · 2025-11-20
>
> **(W3) More evaluation setting.**
>
> **Our evaluation setup follows established SSFL practices** (e.g., SemiFL, FL²), where label shift is the primary heterogeneity benchmark and feature shift is treated as a separate axis. For fair comparison, we therefore report label-shift results in the main paper. To demonstrate robustness beyond this standard setting, **we further evaluate FedOpenMatch under mixed-shift conditions where label and feature shifts coexist, as shown in Fig.13 (revised version). Across all such scenarios, FedOpenMatch consistently outperforms baselines, showing strong resilience to substantial distributional heterogeneity.**
>
> Regarding model scale, the use of ResNet-18 reflects practical FL constraints, as prior works overwhelmingly employ lightweight models. To verify architecture-agnosticity, **we also evaluated FedOpenMatch on WideResNet-28×2, a widely used backbone in SSL and SSFL. FedOpenMatch again achieves clear gains over all baselines, confirming that its effectiveness is not tied to a specific model.**  The resutls are:
>
> |  CIFAR-100    | 50@25@Dir(0.1) | 50@25@Dir(0.1) | 80@25@Dir(0.1)| 80@25@Dir(0.1) |
> | ----- | :-----: | :-----: | :-----: | :-----: |
> |  | Open-set Acc | Closed-set Acc | Open-set Acc | Closed-set Acc |
> | FedLabel | None | 47.12 | None  | 37.05        |
> | IOMatch | 35.71 | 44.58 | 24.65       | 37.12        |
> | FedOpenMatch | 50.16 | 52.14 | 42.69 | 43.23   |
>
> We also appreciate the reviewer’s point on pseudo-labeling. Unlike centralized OSSL methods that update pseudo-labels continuously during local training, FedOpenMatch adopts a round-level update: pseudo-labels are refreshed once per round using the global model and then fixed. Combined with the proposed components, **FedOpenMatch yields higher pseudo-label accuracy and better data utilization as shown in Fig. 9 and 10,** directly contributing to the method’s stability under non-IID drift.
>
> Finally, grouping all unseen classes into a single “unknown” category follows the standard open-set recognition definition, consistent with prior OSSL methods such as OpenMatch and IOMatch.

---

> > ### Author Response · Authors · 2025-11-28
> > **Follow-up on Rebuttal Clarification**
> >
> > Dear Reviewer 5NdG,
> >
> > I hope this message finds you well.
> >
> > We would like to kindly follow up regarding our rebuttal to ensure that all clarifications are clear and sufficient. If there are any remaining questions or points that would benefit from additional explanation, we would be more than happy to provide further details.
> >
> > Thank you very much for your time and consideration.
> >
> > Best regards,
> > The Authors

---

> ### Author Response · Authors · 2025-11-20
> **Looking forward to your further comments**
>
> Dear Reviewer  5NdG,
>
> Thanks a lot for your comments. We've submitted our responses to your concerns on our work. Would you please have a look at our responses and feedback your further comments?
>
> Thanks again!
>
> Best regards,
>
> The authors

---

### Official Review · Reviewer_ikNd · 2025-10-30

**Soundness:** 3
**Presentation:** 3
**Contribution:** 3
**Rating:** 6
**Confidence:** 3

**Summary:**

This paper defines Open-set Semi-Supervised Federated Learning (OSSFL) (unlabeled client data contain unseen classes) and presents FedOpenMatch, the first framework for this setting. It combines an OVA outlier detector with logit adjustment for imbalance, a gradient-stop to decouple OVA/inlier heads, and logit consistency regularization. Experiments on standard benchmarks show sizable improvements in open-set accuracy (e.g., +14.33% on CIFAR-100).

**Strengths:**

- First formalization of OSSFL and a tailored framework
- Strong, consistent gains; well-motivated components (OVA + adjustment + gradient-stop + consistency)
- Good figures/tables and taxonomy (SSFL vs. OSSL vs. FOSR vs. OSSFL).
- Addresses a real-world failure mode for SSFL; likely baseline for future work.

**Weaknesses:**

- Limited analysis of non-IID client shifts (class/feature distribution) on OVA calibration.
- Communication/compute overhead of OVA and consistency losses isn’t profiled; end-to-end systems analysis would help.
- Robustness to extreme open-set ratios and ablations isolating each component could be expanded.

**Questions:**

- How sensitive is OVA performance to class imbalance and thresholds under client heterogeneity?
- Can gradient-stop harm representation sharing between inlier/OVA? Any alternatives (e.g., orthogonal heads)?
- What happens when unknown prevalence is very high or client pools have disjoint seen sets?

---

> ### Author Response · Authors · 2025-11-20
>
> Thanks a lot for your comments. Following are our responses in detail.
>
> **(Q1) Sensitivity to class imbalance and threshold.**
>
> OVA classifiers are inherently sensitive to class imbalance, as each binary classifier treats all non-target classes as negatives. This motivates our use of logit adjustment. Ablation studies show that varying the logit-adjustment weight directly impacts OVA performance under different imbalance levels. Furthermore, the server can estimate the imbalance ratio accordingly, making the method practical for real-world imbalanced datasets.
> To analyse the sensitivity of FedOpenMatch to different OVA thresholds, we performed additional experiments by sweeping the OVA threshold in Fig.12 (revised version). **The results indicate that closed-set accuracy remains stable across thresholds, whereas open-set accuracy fluctuates substantially when the threshold is small. Therefore, using a higher threshold is recommended.**
>
> **(Q2) Can gradient-stop harm representation sharing ?**
>
> As our analysis shows, when the gradient similarity between the two branches is high, training without GS can sometimes perform slightly better, indicating that shared representations are not inherently harmful. However, we observe that the gradient similarity between the inlier and OVA classifiers fluctuates significantly during training, which degrades performance. In contrast, **GS offers a conservative approach that stabilizes training across all stages.** While alternatives like orthogonal heads could also mitigate gradient conflicts, they introduce additional complexity. GS provides a simple solution, and our experiments demonstrate consistent performance improvements across settings.
>
> **(Q3) robustness to high unknown prevalence and disjoint class space.**
>
> To evaluate an extreme scenario with a high prevalence of unknown classes, we conducted experiments on CIFAR-100 using only 20 seen classes in Fig.14 (revised version). Even under this highly challenging setting, FedOpenMatch significantly outperforms baselines, demonstrating its effectiveness when unknown classes dominate the data distribution.
>
> In the Dir(0.1) setting of our main experiments, client label spaces are disjoint, meaning different clients may observe entirely different subsets of seen classes. Across all such configurations, FedOpenMatch consistently outperforms prior methods, confirming its robustness to extreme heterogeneity.

---

> > ### Author Response · Authors · 2025-11-20
> >
> > **(W1) OVA calibration.**
> >
> > Thank you for the suggestion. We agree that non-IID client shifts impose significant challenges for calibrating both the OVA and inlier classifiers. As reflected in our main results, when all other settings are kept constant, Dir(0.1) yields lower performance than Dir(0.3), consistent with the well-known effect that stronger client heterogeneity degrades the global model that is responsible for generating pseudo-labels. This degradation naturally leads to lower pseudo-label quality and thus poorer OVA calibration.
> >
> > In FedOpenMatch, we mitigate this issue by using a high-confidence threshold and a conservative pseudo-labeling strategy, which safeguard pseudo-label quality and reduce the impact of noisy predictions under strong non-IID shifts.
> >
> >
> >
> > **(W2) Communication/compute overhead.**
> >
> > Thank you for pointing this out. We clarify that OVA classifier does not introduce much additional communication and computation overhead as it only consists of a two-layer projection head and a linear classification head. The number of computation and communication overhead are negligible relative to the backbone.
> >
> > **(W3) Ablations.**
> >
> > Thanks for your suggestion. We add additional ablation studies on each component. Here are the results:
> >
> > | CIFAR100@80@25@Dir(0.1) |	Base|	Base+GS |	Base + GS + LA	 |  Base + GS + LA + LCR|
> > | ----- | :-----: | :-----: | :-----: | :-----:|
> > |Open-set Accuracy | 34.81 | 39.51 | 43.65 | 48.36 |
> > |Closed-set Accuracy | 45.86 | 46.14 | 45.81 | 50.38 |
> >
> > The results suggest the effectiveness and necessity of each proposed component.

---

> ### Comment · Reviewer_ikNd · 2025-11-26
>
> Thank you to the authors for the detailed response, which has addressed most of my concerns. I will keep my current score.

---

> ### Author Response · Authors · 2025-11-27
>
> Dear Reviewer iKNd,
>
> Thank you for your timely response. We are pleased that our clarifications have addressed your concerns, and we sincerely appreciate your thoughtful evaluation and recognition of our work.
>
> With our best regards,
>
> The Authors

---

### Official Review · Reviewer_jhAD · 2025-11-01

**Soundness:** 3
**Presentation:** 3
**Contribution:** 3
**Rating:** 6
**Confidence:** 2

**Summary:**

The paper introduces **FedOpenMatch**, the first framework for **Open-Set Semi-Supervised Federated Learning (OSSFL)**—a realistic yet previously unexplored setting where unlabeled client data contain **outliers**. Unlike standard SSFL methods that assume closed-set label spaces, FedOpenMatch jointly trains an **inlier classifier** and a **one-vs-all (OVA) outlier detector** to safely leverage open-set unlabeled data.  Extensive experiments on CIFAR-10, CIFAR-100, and SVHN show FedOpenMatch significantly boosts.

**Strengths:**

The paper demonstrates good **originality** by formally defining a new and realistic problem setting—Open-Set Semi-Supervised Federated Learning (OSSFL)—which bridges the gap between open-set semi-supervised learning and federated learning. The experimental design is also **rigorous**, covering multiple datasets, label scarcity regimes, and heterogeneity levels, with thorough ablation studies validating each component. The writing is clear and well-structured.

**Weaknesses:**

1. The method assumes the server’s labeled set is perfectly balanced and clean, which may not hold in real-world label-at-server settings; robustness to label noise or skewed class priors is unexamined.

2. Second, all experiments use ResNet-18 and synthetic Dirichlet splits—evaluations on larger models (e.g., ViTs) may change the conclusion on feature conflicts between the OVA classifier and the inlier classifier.

**Questions:**

Does your method work with bigger models or more realistic data splits?

All experiments use ResNet-18 and synthetic data splits (Dirichlet). But real-world data may have domain shifts (e.g., different hospitals or cameras), and people now often use Vision Transformers (ViTs).
- Have you tried FedOpenMatch with a ViT or a more realistic non-IID split (e.g., by domain or semantic group)?
- Would the “gradient stop” still be needed in those cases, or is it only helpful for ResNet-18?

---

> ### Author Response · Authors · 2025-11-20
>
> Thank you very much for your comments. We have the responses to your comments in detail as follows:
>
> **(W1). Problem setting.**
>
> First of all, we thank the reviewer for raising this important point. **Our assumption of a balanced, clean labeled set at the server follows standard practice in prior SSFL works**, where the server maintains a small, high-quality labeled dataset while clients hold large-scale unlabeled data. In practical scenarios, the server — typically a company deploying FL— usually has sufficient resources to curate such a modest labeled set, as few as 10–25 samples per class in our setup.
>
> **Regarding robustness to imbalance**: FedOpenMatch does not rely on this balanced assumption. If the server’s labeled data are imbalanced, the imbalance ratio can be estimated from the server’s label distribution and directly applied in Eq. (6) without modifying the framework.
>
> **(W2, Q1). Other models and realistic data split.**
>
> **Evaluation on other models.** Due to limited client resources, most FL works adopt lightweight CNNs, VGG, or ResNet variants [1], and we follow this routine by using ResNet18. Importantly, FedOpenMatch is architecture-agnostic: all components—including gradient stop, logit adjustment, and logit-level consistency—operate on logits or feature gradients and do not rely on model-specific assumptions. To verify this, we conducted additional experiments on CIFAR-100 with **WideResNet-28×2**, following prior SSL and SSFL practices [2,3]. The results are:
>
> |  CIFAR-100    | 50@25@Dir(0.1) | 50@25@Dir(0.1) | 80@25@Dir(0.1)| 80@25@Dir(0.1) |
> | ----- | :-----: | :-----: | :-----: | :-----: |
> |  | Open-set Acc | Closed-set Acc | Open-set Acc | Closed-set Acc |
> | FedLabel | None | 47.12 | None  | 37.05        |
> | IOMatch | 35.71 | 44.58 | 24.65       | 37.12        |
> | FedOpenMatch | 50.16 | 52.14 | 42.69 | 43.23   |
>
> **FedOpenMatch consistently outperforms baselines with this backbone, showing that it scales effectively to different architectures.**
>
> **Evaluation on more realistic data splits.**  Traditional FL studies typically treat label shift and feature shift separately, and prior SSFL works mainly focus on label-shift scenarios. For consistency, our main experiments follow this standard. To further evaluate FedOpenMatch, we designed a more challenging setting **where label shift (via Dirichlet-sampled non-IID labels) and feature shift (heterogeneous visual characteristics) coexist**, see Fig.13 (revised version). Even under these conditions, FedOpenMatch consistently outperforms baselines by a clear margin, demonstrating strong robustness to substantial distributional heterogeneity.
>
> [1] Chen H, Xu T, Wu X, et al. Hybrid Batch Normalisation: Resolving the Dilemma of Batch Normalisation in Federated Learning[C]//Forty-second International Conference on Machine Learning.
>
> [2] Lee S, Le T L V, Shin J, et al. $^ 2$: Overcoming Few Labels in Federated Semi-Supervised Learning[J]. Advances in Neural Information Processing Systems, 2024, 37: 43693-43714.
>
> [3] Huang Z, Shen L, Yu J, et al. Flatmatch: Bridging labeled data and unlabeled data with cross-sharpness for semi-supervised learning[J]. Advances in neural information processing systems, 2023, 36: 18474-18494.
>
> **(Q2). Generalization of gradient stop.**
>
> The gradient stop (GS) is designed to resolve optimization conflicts between the inlier and OVA classifiers. When both classifiers share the same feature extractor, their gradients often point in inconsistent directions, leading to interference and unstable training. This conflict is architecture-agnostic, as it arises from shared feature representations rather than the backbone itself. Accordingly, GS operates independently of the model architecture. **The results in (W2) show comsistent performance gains among baselines when employing different model.**
>
> We further compare the performance of FedOpenMatch with and without GS and plot the gradient similarity between classifiers when employing  WideResNet-28×2 as backbone. The results are presented in Fig. 15 (revised version), showing **the gradient similarity remains low during the whole training process, and the GS improves the performance effectively.**

---

> > ### Comment · Reviewer_jhAD · 2025-11-28
> >
> > Thanks for your response — I’ll take your feedback into consideration when rating.

---

> > > ### Author Response · Authors · 2025-11-28
> > >
> > > Dear Reviewer jhAD,
> > >
> > > Thanks a lot for your time and consideration. Please feel free to let us know if any additional clarification would be helpful—we would be glad to provide further details.
> > >
> > > Best wishes,
> > >
> > > The Authors

---

### Author Response · Authors · 2025-12-02
**Summary of the Discussion for the Area Chair**

We sincerely thank all the reviewers for their constructive comments and helpful discussion. We also appreciate the Area Chair’s time and effort to examine both the reviewers’ assessments and our responses for reaching a fair and well-informed decision. To facilitate the Area Chair’s evaluation, we **summarize the key points of our responses to the reviewers’ comments and reviewers’ feedback** as follows:

**1. About the review of Reviewer jhAD (rating 6; S/he checked our rebuttal, no further questions)**

Reviewer jhAD acknowledges the originality of the proposed OSSFL problem and framework, the rigor of our experimental design, the thoroughness of the ablation studies, and the clarity and structure of the manuscript.

In our rebuttal, we addressed all her/his concerns by providing additional experiments to demonstrate the robustness of FedOpenMatch across different backbone and more complex data settings, and empirically confirming the model-agnostic nature of the Gradient Stop mechanism. Regarding the problem formulation, we clarified that our setup strictly follows the standard conventions established in prior SSFL literature.

Reviewer jhAD replied to our rebuttal and promised to take our feedback into her/his final rating.

**2. About the review of Reviewer ikNd (rating 6; acknowledged that our responses addressed most of her/his concerns)**

Reviewer ikNd recognizes the originality of the OSSFL formulation, the clear motivation behind each component, the strong and consistent performance improvements, and the quality of the figures, tables, and taxonomy.

In response to her/his concerns, we conducted additional experiments evaluating different OVA thresholds, extreme open-set ratios, and further ablations isolating each component. We also provided supplementary analysis of the communication and computation overhead of the OVA classifier and clarified the motivation and effectiveness of the Gradient Stop mechanism.

Reviewer ikNd checked our rebuttal, and acknowledged that our responses addressed most of her/his concerns, and determined to keep her/his original rating of 6.

**3. About the review of Reviewer 5NdG (rating 4; no any feedback in the discussion period)**

Reviewer 5NdG acknowledges the clarity of the problem definition and the performance gains. We conducted additional experiments using a different backbone and more complex data settings, which directly address the reviewer’s concerns regarding evaluation setup.

We must point out that **most of this reviewer’s concerns arise from her/his misunderstanding of the problem setting**. Our work studies open-set SSFL, which is fundamentally different from open-world SSFL in both task definition and assumptions. Prior open-world SSFL follows the label-at-client paradigm (e.g., requiring 50% labeled client data), whereas our method follows the label-at-server configuration, the standard and realistic setup in SSFL. Moreover, grouping unseen classes into a single “unknown” category is consistent with the long-standing convention in OSSL.

As for baseline strength, in our paper all OSSL baselines strictly follow their official implementations and the recommended hyperparameters from prior SSFL works; no shared hyperparameters were tuned in our favor. All experiments use identical random seeds (0,10,100), and full source code is provided. The instability observed in these baselines is inherent to their design under federated conditions, not an artifact of our implementation.

Unfortunately, Reviewer 5NdG didn’t provide any feedback during the discussion period (before Nov. 28).

**4. About the review of Reviewer abnT (rating 6; acknowledged that our responses addressed most of her/his concerns)**

Reviewer abnT recognizes that FedOpenMatch is practically valuable, well-justified through solid experiments, supported by sufficient ablation studies, and clearly written.

To address her/his concerns, we conducted additional experiments with varying local epochs and a different backbone to further clarify the behavior of Gradient Stop. The experimental results demonstrate the effectiveness of the proposed method.

Regarding novelty, we reinstated that FedOpenMatch is not a trivial combination of prior techniques. Guided by Occam’s Razor, the method is intentionally simple yet highly effective. Each component is introduced to address a specific OSSFL failure mode, and this principled design offers practical benefits—ease of implementation, interpretability, and strong baseline value for future research.

Reviewer abnT checked our rebuttal, and acknowledged that our responses addressed most of her/his concerns, and determined to keep her/his original rating of 6.

---

### Meta-Review · Area_Chair_97Ze · 2025-12-05

**Summary:**

The paper introduces the open-set semi-supervised federated learning (OSSFL) setting and a simple, principled framework (FedOpenMatch) that consistently improves open- and closed-set performance across datasets and heterogeneity regimes. Reviewers highlight strong experimental coverage, ablations, and practical value; the rebuttal further strengthens the case with mixed-shift and alternative-backbone results, threshold/ratio sensitivity, and per-component studies while clarifying compute/comm overheads. Remaining concerns include perceived incremental novelty, missing empirical tests for noisy/imbalanced server labels, and limited systems profiling/gradient stop alternatives. Given that the core problem is timely and important in FL, the method is effective and reproducible, and most substantive concerns were addressed or reasonably mitigated, I recommend "accept (poster)".

**Reviewer Concerns:**

**jhAD** Concerns: balanced/clean server labels; reliance on ResNet-18 & Dirichlet splits; whether Gradient-Stop is architecture-agnostic. Addressed: authors added WideResNet and mixed-shift experiments; argued GS/model-agnosticism; reviewer acknowledged. Outstanding: no empirical test of label noise or skewed server priors.

**ikNd** Concerns: OVA calibration under heterogeneity; overhead of OVA/consistency; robustness at extreme open-set ratios; deeper ablations. Addressed: threshold sweeps, extreme-unknown setting, per-component ablations, note that OVA head is lightweight; reviewer kept score. Outstanding: fuller end-to-end systems profiling; alternatives to GS (e.g., orthogonal heads) unexplored.

**5NdG** Concerns: novelty vs. open-world SSFL; baseline strength; narrow evaluation; "unknown" grouping. Addressed: detailed clarifications, mixed-shift/WideResNet results, round-level pseudo-labels. Outstanding: novelty/baseline concerns (no discussion reply).

**abnT** Concerns: method combines existing ideas; sensitivity to local epochs; gradient-similarity explanation; equation sign. Addressed: added local-epoch study; clarified logit-adjustment form; novelty perception remains.

**Reviewer Scores:**

**jhAD** Rebuttal likely sustains 6.

**ikNd** Explicitly kept score 6 after rebuttal.

**5NdG** Clarifications may nudge to 6 (not sure).

**abnT** Acknowledged responses; maintained 6 due to perceived limited novelty.

---

### Decision · Program_Chairs · 2026-01-26

Accept (Poster)